# Drug Reaction with Eosinophilia and Systemic Symptoms (DRESS): Focus on the Pathophysiological and Diagnostic Role of Viruses

**DOI:** 10.3390/microorganisms11020346

**Published:** 2023-01-30

**Authors:** Giuseppe A. Ramirez, Marco Ripa, Samuele Burastero, Giovanni Benanti, Diego Bagnasco, Serena Nannipieri, Roberta Monardo, Giacomo Ponta, Chiara Asperti, Maria Bernadette Cilona, Antonella Castagna, Lorenzo Dagna, Mona-Rita Yacoub

**Affiliations:** 1Unit of Immunology, Rheumatology, Allergy and Rare Diseases, IRCCS Ospedale San Raffaele, 20132 Milan, Italy; 2Faculty of Medicine, Università Vita-Salute San Raffaele, 20132 Milan, Italy; 3Unit of Infectious Diseases, IRCCS Ospedale San Raffaele, 20132 Milan, Italy; 4IRCCS Policlinico San Martino, Department of Internal Medicine (DIMI), University of Genoa, 16132 Genoa, Italy

**Keywords:** DRESS, virus, eosinophils, reaction, T-cells, herpesvirus, viral reactivation

## Abstract

Drug reaction with eosinophilia and systemic symptoms (DRESS) is a heterogeneous, multiorgan and potentially life-threatening drug-hypersensitivity reaction (DHR) that occurs several days or weeks after drug initiation or discontinuation. DHRs constitute an emerging issue for public health, due to population aging, growing multi-organ morbidity, and subsequent enhanced drug prescriptions. DRESS has more consistently been associated with anticonvulsants, allopurinol and antibiotics, such as sulphonamides and vancomycin, although new drugs are increasingly reported as culprit agents. Reactivation of latent infectious agents such as viruses (especially Herpesviridae) plays a key role in prompting and sustaining aberrant T-cell and eosinophil responses to drugs and pathogens, ultimately causing organ damage. However, the boundaries of the impact of viral agents in the pathophysiology of DRESS are still ill-defined. Along with growing awareness of the multifaceted aspects of immune perturbation caused by severe acute respiratory syndrome coronavirus 2 (SARS-CoV-2) during the ongoing SARS-CoV-2-related disease (COVID-19) pandemic, novel interest has been sparked towards DRESS and the potential interactions among antiviral and anti-drug inflammatory responses. In this review, we summarised the most recent evidence on pathophysiological mechanisms, diagnostic approaches, and clinical management of DRESS with the aim of increasing awareness on this syndrome and possibly suggesting clues for future research in this field.

## 1. Introduction

Adverse drug reactions constitute an emerging issue for public health due to their increasing incidence over time, at least in Westernised countries [1,2]. Population aging along with the growing prevalence of dementia and multi-organ morbidity are associated with frequent institutionalisation and enhanced drug prescribing and may account for this trend [3]. Hospitalisation-related adverse drug reactions are particularly relevant from an epidemiological and economic standpoint and can occur in up to 30% of patients entering ordinary wards [4]. Drug-related hypersensitivity reactions (DHR) constitute a subgroup of adverse drug reactions, occurring with an incidence rate of 80 cases/1000 person-years in in-patient settings [5]. DHRs pose major challenges to the management of hospitalised patients, since they occur unpredictably (in contrast with non-immune-mediated adverse drug reactions) and affect patients' ability to receive appropriate treatments for their acute conditions, besides bearing intrinsic morbidity and mortality risks [6]. Multiple pathogenic factors are thought to contribute to the development and maintenance of hypersensitivity reactions to drugs. Aberrant haptenation of drugs in the setting of acute inflammation and/or direct drug-related activation of immune cells might combine with predisposing genetic factors, such as a permissive human leukocyte antigen (HLA) repertoire, to prompt drug sensitisation [7,8,9,10,11,12,13]. Microbial factors might also contribute to the development of DHRs by promoting systemic inflammation and affecting immune tolerance due to molecular mimicry among self and microbial antigens. Consistently, the majority of DHRs are associated with the use of antimicrobials [5].

Drug reaction with eosinophilia and systemic symptoms (DRESS, also known as drug-induced hypersensitivity syndrome, DIHS) is a rare but potentially life-threatening delayed-type systemic DHR characterised by elevated blood eosinophil counts along with constitutional symptoms and multi-organ failure [14,15]. Affordable estimates of DRESS epidemiology are to date missing. The current literature suggests that DRESS incidence can range from less than 0.01 cases to 0.7 cases per 1000 hospitalised patients depending on the healthcare system and demographic context [16,17,18,19]. Changing epidemiological trends within the same cohort according to variations in drug prescription attitudes and environmental factors have also been reported [20]. Infectious agents including exogenous or latent virus along with endogenous retroviral elements constitute known perturbators of the physiological immune response and are increasingly recognised as cofactors in the onset of DRESS. This review summarises the clinical and pathophysiological evidence addressing the role of microbial cofactors in DRESS up to date.

## 2. Aetiology and Pathogenesis

DRESS is a systemic disorder sustained by two pairs of fundamental pathophysiological pillars: (1) inciting stimuli, encompassing drugs; and viruses; (2) deranged immune responses including (a) HLA-restricted aberrant T-cell activation and; (b) eosinophilic inflammation (Figure 1).

### 2.1. Drugs

A straightforward association with drug exposure is found in 80% of patients, and DRESS onset typically occurs 2–8 weeks after treatment start with the causative drug [21]. Recent drug discontinuation is also associated with DRESS. Numerous drugs have been described as possible triggers of DRESS, but around 75% of cases can be traced back to a high-risk group of drugs [19,22], including anticonvulsants, allopurinol, antibiotics such as vancomycin, minocycline, trimethoprim-sulfamethoxazole and other sulphonamides, antituberculosis agents and antiviral drugs such as nevirapine [23]. Shorter lag times for DRESS onset are observed when antibiotics or iodinated contrast media are implicated [24,25].

However, other drugs have also been reported in association with DRESS onset. Some of these drugs are in more widespread use, such as anti-inflammatory drugs (e.g., non-steroidal anti-inflammatory drugs, NSAIDs, and paracetamol) or antipsychotic drugs. Special attention should be given to special populations such as patients with cancer, rheumatic diseases and chronic viral infection. These patients bear disease-specific risk factors for developing DRESS, including concomitant polytherapy, concomitant infection, kidney or liver failure, and intrinsic immune dysfunction [23]. Furthermore, novel drugs are continuously introduced in the setting of these disorders, expanding the spectrum of potential DRESS triggers. Anti-cancer targeted therapies such as imatinib [21] or sorafenib [22], immune-modulators such as IL1 or IL6 inhibitors (e.g., anakinra, canakinumab, tocilizumab or hydroxychloroquine) or anti-HCV therapies including Telaprevir and Boceprevir constitute potential examples of emerging culprit drugs for DRESS (Table 1) [26].

Culprit drug cross-reactivity is not conventionally expected in DRESS. However, very limited evidence suggests that sensitisation to glycopeptides and *β*-lactams might compromise patient ability to eventually receive any members of these drug classes [27]. In addition, patients with DRESS may develop secondary neosensitisation to unrelated chemical compounds concurrently administered during DRESS [28].

From a pathophysiological standpoint, trigger drugs are thought to constitute the main target of the immune response. The strength of association between drugs and DRESS is affected by interindividual and inter-ethnic variations in the HLA repertoire (see below). Distinct HLA variants might in fact segregate with selected ethnicities. Additional inherited factors may promote altered drug metabolism and variably combine with HLA-related factors to contribute to DRESS susceptibility. Polymorphisms in cytochrome P (CYP) 450 and N-acetyltransferase (NAT1, NAT2) may affect drug pharmacokinetics and cause active ingredients of metabolism by-product overload [29]. Examples of the roles of these polymorphisms are constituted by the associations between CYP2C9*3 and severe reactions to phenytoin (in Asian ethnicities) and between variants of the NAT gene and sulphonamides [30,31]. This evidence raises the possibility that non-immunological, non-virological factors impacting drug metabolism may play a role on DRESS.

Besides constituting a target for deranged immune response, culprit drugs might also be involved in other disease mechanisms, including viral reactivation. For example, continuous anticonvulsant therapy has been shown to associate with IgG production decreases. Impaired humoral immunity in turn constitutes a risk factor for viral reactivation (see below).

**Table 1 microorganisms-11-00346-t001:** Most frequent and newly reported culprit drugs in DRESS.

Drugs Categories	Drug		Ref.
Urate lowering agents	Allopurinol	High Risk	[32]
	Febuxostat		[33]
Aromatic antiepileptic agents	Carbamazepine	High Risk	[14]
	Phenytoin	High Risk	
	Lamotrigine	High Risk	
	Oxcarbazepine	High Risk	
	Phenobarbital	High Risk	
Sulphonamides	Sulfasalazine	High Risk	[14]
	Dapsone	High Risk	
	Trimethoprim-Sulfamethoxazole	High Risk	
	Sulfadiazine	High Risk	
Antibiotics	Vancomycin	High Risk	[23]
	Minocycline	High Risk	
	Piperacillin-Tazobactam		
	Antituberculosis Agents	High Risk	
	Other Penicillins and Cephalosporins		
Antiviral agents	Nevirapine	High Risk	[34]
	Abacavir	High Risk	[35]
	Efavirenz		[36]
	Boceprevir		[37]
	Telaprevir		[38]
Anti-inflammatory drugs	Diclofenac		[23]
	Celecoxib		
	Ibuprofen		
Anti-IL1 antibodies	Anakinra		[39]
	Canakinumab		
Anti IL6 antibodies	Tocilizumab		[39]
Targeted therapies	Imatinib		[21]
	Sorafenib		[40]
	Vismodegib		[41]
	Vemurafenib		[42]
Antipsychotic agents	Fluoxetine		[23]
	Olanzapine		
Anti-coagulant	Rivaroxaban		[43]
Immunomodulators	Hydroxychloroquine		[44]

### 2.2. Viral Factors

Clinically, viral reactivation occurs up to two weeks after the onset of DRESS symptoms and is associated with a worse prognosis in term of disease duration, relapses, constitutional symptoms and organ involvement [45,46,47], as compared to patients with no evidence of viral reactivation [47,48]. The pathophysiological meaning of viral reactivation in DRESS and the reciprocal interactions between viral reactivation and inflammation are the subjects of an ongoing debate [49]. Viral reactivation may take part in DRESS pathogenesis in four ways (Figure 1):Viruses may cause direct tissue damage and contribute to the early manifestations of DRESS.In a later phase of the disease course, they can be the target of the immune response [50,51]. In this regard, an “immune-reconstitution like” syndrome may occur as the result of corticosteroid treatment and/or immunosuppression to control DRESS manifestations [48].Viral reactivation might constitute the epiphenomenon of a wider expansion of virus-harbouring immune cells in the setting of systemic inflammation. In fact, latent human herpesviruses (HHVs) chronically resides in cells of the immune system, including T-lymphocytes and cells of the monocyte/macrophage lineage. Thus, viral reactivation and release could represent an early marker of stimulation of these cell reservoirs following drug-driven expansion, rather than representing a trigger event of DRESS [52].Viruses might promote anti-drug responses and mis-differentiation of antigen-specific lymphocytes by molecular mimicry. In fact, T-lymphocytes previously selected and expanded by viral antigens might eventually be activated by drugs, inducing DRESS (see also below at Section 2.3.2) [53]. Furthermore, challenging EBV-immortalised B-lymphocytes from healthy subjects and from patients with DRESS with DRESS culprit drugs selectively prompts EBV production increases in DRESS subjects [54], suggesting generalised dysfunction of tolerance and pathogen control in both arms of the immune response during DRESS.

Therefore, DRESS pathophysiological unicity may reside in the feed forward loop linking drug-induced triggering of memory lymphocytes followed by additional antigenic stimulation due to viral reactivation.

#### 2.2.1. Human Herpesviruses (HHVs)

A typical feature of DIHS/DRESS is the reactivation of latent HHVs, namely HHV-6, HHV-7, Epstein–Barr virus (EBV), and Cytomegalovirus (CMV) [22,49,55,56,57,58,59,60,61,62,63,64,65,66]. Herpes viruses are known to promote the reactivation of other viruses upon reactivating themselves with a peculiar reactivation sequency, as originally reported by Kano 2006 [63] and recently reviewed by Anci 2021 [58]. As suggested by the literature, herpesviruses have differential reactivation kinetics, which can also be observed in the same subjects. The first viruses to reactivate are HHV-6 and EBV, followed by HHV-7 and finally CMV, in the same order that occurs in graft versus host disease (GVHD).

Sequential herpesvirus reactivation can also account for the long-lasting clinical picture of DIHS/DRESS and the occurrence of delayed organ complications even after discontinuation of culprit drugs [46,49,56,57,63,67]. This evidence also suggests that viral reactivation itself is presumably not involved in the onset of DRESS but may be a crucial factor determining the prolonged clinical course of this condition [29,58,65,68]. Furthermore, the characteristic 20%-mortality risk of DRESS/DIHS mortality is significantly affected by CMV reactivation besides older age, hepatic and kidney involvement, while EBV reactivation is most often observed in patients with milder DRESS presentations [29,55,69].

HHV-6 positivity also appears to be associated with a more severe disease course and a later onset following drug exposure than in the case of HHV-6-negative DRESS [48,51,58]. HHV-6 is the most frequent HHV to be associated with DRESS, and its reactivation typically occurs during the course of DRESS and up to 2–3 weeks after DIHS/DRESS rash onset [46,49,67]. HHV-6 infects the vast majority of the general population during infancy and has been shown to be able to be chromosomally integrated into host DNA both in the general population and in the setting of DIHS/DRESS [29,70,71,72]. Conversely, DIHS/DRESS has rarely been reported in patients during primary viral infection [73,74]. HHV-6 reactivation is normally a transitory event; nevertheless, in some cases, the virus can be detected also several weeks after the onset of DRESS, leading generally to the recurrence of skin rash [49,75,76]. During the course of DRESS, HHV-6 DNA can be found in the skin, lymph nodes, kidney, and liver, along with detection of HHV-6-derived microRNAs in serum and circulating mononuclear cells, suggesting the potential concurrent role of reactivated HHV-6 in the development of DRESS-related rash, lymphadenopathy, and organ failure [49,64,77,78,79]. The detection of HHV-6 DNA was also associated with symptom flare-ups, and the increase in HHV-6 DNA levels correlated well with the severities of the inflammatory responses [49,57,70,79,80].

HHV-7 has also been demonstrated to reactivate in patients with DRESS, albeit its clinical impact is not fully elucidated. Notably, two separate prospective studies systemically evaluated the proportion of DRESS patients with HHV-7 reactivation. In a study by Picard et al., 32% of patients demonstrated HHV-7 reactivation (compared to 45% of subjects with HHV-6 reactivation) [54]. On the other hand, in another study by Chen et al., only 1/23 patients experienced HHV-7 reactivation [66]. As the studies were performed in different geographical settings, it is possible that this striking difference in the proportion of patients with HHV-7 reactivation could be more related to epidemiological factors than to actual pathophysiological mechanisms.

Among other human herpesviruses, herpes simplex virus (HSV) reactivation has rarely been reported, and usually occurs early during the course of the disease followed by a rapid reduction of HSV-DNA titres [58,81]. Few reports of complications due to reactivation of HSV or VZV in DIHS/DRESS have been published [81,82]. By contrast CMV reactivation can induce some of the late-onset complications of DIHS/DRESS [49,59,83], which can present up to two months after the onset of DIHS/DRESS and even culminate with death. This occurs especially in the case of evident CMV disease, whose manifestations can be hepatitis, pneumonia, gastroenteritis, and skin and gastrointestinal ulcers. Indeed, the higher mortality described in patients with DIHS/DRESS with CMV reactivation may, at least in part, be related to viral end-organ disease, which occurs more commonly in the case of CMV reactivation compared to other viruses. Nonetheless, the association between CMV reactivation and mortality in critically ill patients has been described in multiple studies (recently summarized by Lachance et al. [84] and Schildermans et al. [85]), even though the underlying physiopathological mechanisms are not entirely understood. Interestingly, studies analysing the use of antivirals for prophylaxis of CMV reactivation in critically ill patients failed to show a benefit of this intervention [86,87,88], and, therefore, this approach should be discouraged in patients with DRESS, while treatment of end-organ disease should be promptly instated.

#### 2.2.2. Severe Acute Respiratory Syndrome Coronavirus 2 (SARS-CoV-2)

DIHS/DRESS has been described among patients with SARS-CoV-2-related disease (COVID-19). These patients are particularly prone to classical risk factors for DIHS/DRESS, as they are frequently affected by multiple comorbidities and exposed to several drugs potentially associated with this syndrome. Few isolated cases of DIHS/DRESS in patients with COVID-19 have been described in the literature [89,90,91,92,93,94,95,96,97,98], but the actual incidence of this condition in patients with SARS-CoV-2 pneumonia is still to be fully explored. In a retrospective cohort analysing 9330 patients hospitalised with COVID-19 from a US healthcare system [99] between January 2020 and May 2021, six cases of DRESS syndrome were identified, corresponding to an incidence of 6.43 per 10,000 patients. The most likely culprit drugs were antibiotics, which were administered to all patients who developed DRESS (specifically vancomycin in 6/6, cefepime in 4/6, and meropenem in 1/6 patients). Nevertheless, all antibiotics were prescribed empirically, without microbiological evidence of a secondary bacterial infection. Interestingly, all patients in the study had markedly high eosinophilia (>3.00 × 10^6^ cells/L), and no deaths were reported. In another cohort study [20], five cases of DRESS syndrome were identified among 2721 patients admitted with COVID-19 between February 2020 and March 2021. Notably, all cases were identified during the first wave (February to May 2020), with a corresponding incidence rate of 0.17/100 patient-months (compared to 0.0005/100 patient-months recorded in the previous 3 years in the same institution). Hydroxychloroquine (prescribed in 4/5 cases) and *β*-lactam antibiotics (administered to 4/5 patients) were considered the most probable culprit drugs, even though all patients received multiple drugs that could, at least, be considered to have a possible causative role. The authors argued that the differences in incidence of DRESS syndrome between separate COVID-19 waves could have been attributable to the evolution in the management of COVID-19, as patients were less exposed to potential culprit drugs such as hydroxychloroquine or lopinavir/ritonavir (and possibly antibiotics) and more frequently received corticosteroids, which could have dampened the immunological mechanisms leading to DRESS syndrome. Indeed, hydroxychloroquine is an emerging potential culprit drug in the setting of DRESS [14,44,90,94,100] (Table 1).

Interestingly, some authors postulated a possible role of SARS-CoV-2 infection in the pathogenesis of DRESS syndrome. The cytokine storm seen in patients with COVID-19 and maculopapular drug rashes was shown to promote the activation of monocytes/macrophages and a robust cytotoxic CD8+ T-cell response. This immunological profile was seen, to a lesser extent, in non-COVID-19 patients with DRESS syndrome, but not in patients with other maculopapular drug rashes [101]. Specifically, COVID-19 and DRESS patients were shown to share an upregulation of several inflammatory cytokines, namely IL-6, TNF, IL-8, IFN-γ, CXCL9, CXCL10 and CXCL11, accompanied by an increase in IL-4 and IL-5 (representing a type 2 response) and proteins associated with eosinophil chemotaxis and immune suppressive phenotype. Therefore, it is possible to speculate that the T-cell hyperactivation and systemic cytokine storm seen in COVID-19 patients may be a predisposing factor for delayed drug hypersensitivity reactions [102], also given the absence of SARS-CoV-2 RNA in skin biopsies from patients affected by maculopapular skin rashes. Moreover, the reactivation of HHV-6, EBV and CMV has been described in patients with COVID-19 [103,104,105], highlighting a possible shared pathophysiological mechanism.

In conclusion, there is currently a paucity of data regarding the possible relationship between SARS-CoV-2 infection and DRESS, from both the clinical and biological points of view. While some authors described a higher incidence of DRESS in COVID-19 patients compared to historical cohorts, it is plausible that this finding may be related to the presence of several risk factors in these patients, namely a widespread use of antibiotics and other potential culprit drugs. Studies including a similar population (matched for demographic characteristics, comorbidities, and drug prescription) could shed light on this matter. Based on the data currently available, physicians should maintain a high index of suspicion and promptly discontinue potential culprit drugs in patients with COVID-19 with suspected DRESS, even in the absence of an intrinsically higher risk in this population.

#### 2.2.3. Other Viruses

DIHS/DRESS has been described also in association with viruses that do not belong to the Herpesviridae family. In a single case, Coxsackie B4 was reported in a patient who developed a fulminant type 1 diabetes mellitus during DIHS/DRESS induced by carbamazepine. A serological panel was requested and a rise in anti-Coxsackie B4 immunoglobulin titre from <1:4 to 1:64 was observed. However, specific anti-HHV-6 IgG increased by 64-fold, too. Thus, it remains difficult to establish a clear pathogenetic relationship between DRESS and Coxsackie virus [106].

In another report, influenza virus was associated with DIHS/DRESS. A 35-year-old woman with rheumatoid arthritis developed DRESS-related symptoms 6 weeks after starting sulfasalazine. She was tested for serology and antigens of different viruses, and only influenza A and B turned out positive [107]. This is the only DIHS/DRESS case related to influenza virus currently described in the literature.

Another case report describes a young man who developed DIHS/DRESS syndrome and was diagnosed with chikungunya fever (chikungunya IgM titre 1:80 with a reference range of 1:10). As in the previous case, the patient had been treated with sulfasalazine for joint pain in the previous months, making it difficult to link chikungunya virus to DRESS syndrome [108]. Moreover, chikungunya virus-infected patients can develop mucocutaneous changes that may mimic the clinical presentation of DIHS/DRESS, thus making a differential diagnosis more difficult [109,110]. Notably, the macular hyperpigmentation of the nose and cheeks that sometimes follows chikungunya infection (Chik sign) initially appears as a maculopapular exanthem [111] resembling DRESS rash. In both scenarios, activation of skin-resident memory T-cells may account for anti-infectious responses and hypersensitivity reactions, such as DRESS [112]. The pathogenesis of the later hyperpigmentation of chikungunya is unclear, even though some reports in the literature hint at increased intraepidermal melanin dispersion or retention, triggered by chikungunya fever [113].

Finally, another virus that is often cited in relation to DIHS/DRESS is HIV. However, the use of antiretroviral drugs seems to be the trigger, while the virus itself is likely a bystander. An interesting case series describes six patients who developed DIHS/DRESS under treatment with raltegravir for HIV infection; five of these patients were of African ethnicity and four of them possessed the HLA-B*53:01 allele, thus suggesting a possible genetic predisposition for the development of DIHS/DRESS when exposed to raltegravir [114]. Other antiretroviral drugs that have been linked to DIHS/DRESS are nevirapine [34,115] and abacavir in patients who expressed the HLA-B*57-01 allele [35], even though with the latter drug, a hypersensitivity reaction occurs without haematological abnormalities or internal organ involvement. Lastly, a South African case series reported six patients coinfected with HIV and tuberculosis who developed DRESS syndrome after starting rifampicin [116], but even in this case, the link between DIHS/DRESS and HIV was weak, given that antitubercular drugs are often associated with cutaneous adverse drug reactions [117].

#### 2.2.4. Immunological Mechanisms of Virus Reactivation 

Multiple factors coincide with virus reactivation in the setting of DRESS. T-cells and antibodies act synergically against virus dissemination by preventing viral reactivation from a latent stage and by preventing the spread of reactivating lytic virus, respectively. Therefore, DIHS/DRESS may develop at the crossroads between transient humoral adaptive immune dysfunction with decreased B-cell counts and antibody secretion, reactivated HHV and expansion of drug-specific T-cells (see, for instance, Aihara 2003 [118] and Kano 2004 [119]). An early decrease in total IgG levels (mostly observed during the acute phase of DRESS) might also corroborate the clinical suspicion of DRESS and might facilitate HHV-6 reactivation [49,56]. Indeed, viral reactivation may potentially induce a secondary immune response with subsequent increase in the levels of specific anti-HHV-6 IgG, mostly observed in later stages of DRESS. It is also possible that the causative drug may induce a state of immunosuppression, subsequently allowing HHV reactivation [50].

One of the reservoirs of human latent HHV-6 infection is represented by mono/myeloid cells, which appear particularly prone to HHV-6 spreading in patients with DRESS. Specifically, circulating CD11b + CD13 + CD14 − CD16^high^ mono/myeloid precursors rise in the early stage of the disease course. In addition, these cells express high levels of OX40L, promoting interactions with their lymphocytic counterpart, which in turn expresses supranormal levels of the cognate receptor CD134 (=OX40) following systemic activation [49,70,120]. Strikingly, CD134 is also a cell-specific receptor for HHV-6 [49,70]. Furthermore, DRESS cases following immune checkpoint inhibitor exposure are increasingly reported [121,122]. Circulating CD11b + CD13 + CD14 − CD16^high^ mono/myeloid precursor cells harbouring HHV-6 also express a skin-homing molecule, CD194 (=CCR4), and are responsive to high mobility group box (HMGB-1). In the skin and in the blood of patients with DIHS/DRESS, high levels of HMGB-1 have been found. Taken together, these data suggest that HHV-6 reactivation might initiate in the skin [22] and consists in monocytes/macrophages latently infected by HHV-6 reactivating during the early phase of DIHS/DRESS, leading to increased viral loads, and subsequent infection of CD4+ T cells via CD134 [49]. This mechanism might account for the preferential involvement of the skin in the clinical spectrum of DRESS. Consistently, the expression of HHV-6 cellular receptors in skin lesions soon after onset positively correlates with DIHS severity [49,70].

Patients with DIHS/DRESS can present high levels of plasmacytoid dendritic cells (pDCs) in the affected skin regions, but, on the contrary, low levels of pDCs in the peripheral blood. Interferon α (IFNα), produced by pDCs, inhibits viral infection and connects innate and adaptative antiviral immunity. In fact, IFNa triggers the antiviral response of myeloid dendritic cells (mDCs), T-cells and natural killer cells and also the maturation of B-cells in order to promote IgG production for antiviral response. When pDCs migrate from the circulation to the skin, the number of pDCs in blood is reduced, possibly resulting in reduced antiviral responses [123].

The cytokine milieu can also affect viral reactivation. Interestingly, a G-CSF-, MIP-1α-, TNF-α-, IL-8-, IL-10-, IL-12p40-, and IL-15-enhanced profile as observed in DRESS has been shown to be associated with CMV reactivation, and higher eotaxin, IL-10, and G-CSF levels accompanied with lower IL12p40 levels at baseline might be useful for predicting the development of CMV disease [124]. Patients at risk of CMV reactivation can be identified by surveillance of these cytokine/chemokine levels prior to and after beginning immunosuppressive therapy. This may help in preventing morbidity and mortality.

### 2.3. T-Cell Responses

The role of T-cells in DRESS is clinically supported by evidence of positive patch testing and of activation of drug-specific CD4+ and CD8+ T-lymphocytes in patients with DRESS [125,126]. Pharmacogenomic and functional data (see below) point to a prominent role of CD8+ T cells in mediating anti-drug and anti-viral responses along with non-typical support for eosinophil recruitment. Nonetheless, evidence of antigen-specific CD4+ T-cell activation and expansion in DRESS has also consistently been reported [49,125,126]. Drug-reacting cells typically produce large amounts of cytokines potentially associated with a broad spectrum of inflammatory phenotypes that include IL-4, IL-5, IL-13, IFN-γ, and TNF-α [60]. Multiple aspects of T-cell biology may contribute to pathophysiological mechanisms underlying DRESS [22]. Recent evidence indicates that genetically determined dysfunctions in the control of apoptosis and proliferation might contribute to susceptibility to severe cutaneous adverse reactions (including DRESS) in populations of European descendent. Nicoletti et al. performed a recent meta-analysis of two genome-wide association studies (GWAS) on patients with phenotypically defined carbamazepine-serious cutaneous adverse reaction (CBZ-SCAR) and carbamazepine-drug-induced liver injury (CBZ-DILI). They found that an uncommon variant in the ALK gene conferred a supranormal risk of CBZ-SCAR. Indeed, the ALK gene is a receptor tyrosine kinase found in numerous tissues, being involved in cellular proliferation and cell death. This evidence could suggest that the expression of this gene variant may have a relevant role both in T-cell function (as far as proliferation is concerned) and keratinocyte biology (by affecting mechanisms of cell death). These findings also suggest that cellular homeostasis, besides immune-specific functionality, might be altered in T-cells in the setting of DRESS [127].

#### 2.3.1. HLA

HLA-restricted antigen-specific recognition followed by cellular activation constitutes the hallmark of T-cell-mediated responses. HLA is a complex of genes mapping to chromosome 6p21.3 in humans and encoding cell-surface proteins responsible for several activities of the immune system, including self-non-self-recognition and presentation of antigen on the membranes of specialized cells. HLA is highly polymorphic in the human population, and associations between the risk of developing DRESS and several HLA genetic variants have been reported [56,128,129]. These associations are usually drug-specific, possibly implicating that some HLA molecules are able to interact with a specific drug in a more efficacious way to activate T lymphocytes [130]. Although both CD4+ and CD8+ T lymphocytes can be activated by drug exposure in DRESS [125,126], the class I HLA profile shows a stronger epidemiological association with DRESS than the class II HLA profile [55]. Besides the association with the risk of becoming sensitised to selected drugs, HLA is also linked to susceptibility to infection and chronicisation of viral infection. A summary of most frequent DRESS-related HLA variants and their effects on viral infection is reported in Table 2.

Secondary to the prominent pathogenic role of HLA in DRESS and other DHRs, HLA testing has high specificity and negative predictive value for predicting the occurrence of such reactions in patients exposed to known DHR triggers, suggesting its potential clinical use [19]. However, implementation of HLA genotyping into routine clinical practice is mostly affected by the number needed to test (NNT) in order to prevent one case of DHR. In turn, NNT is affected by DHR incidence and HLA frequency in a given population. HLA-B*57:01 screening is part of routine clinical practice for candidates for abacavir, due to the relatively high frequency of abacavir hypersensitivity syndrome in patients treated with abacavir, at least in Caucasians. Similarly, screening for HLA-B*15:02 and HLA-B*58:01 has a low NNT for carbamazepine-related SJS/TEN and for allopurinol-related DHR, respectively, in Asian populations, due to the high frequency of these alleles in these populations. Conversely, some drug regulatory agencies recommend HLA-A*31:01 genotyping for non-Asian patients due to receive carbamazepine. Due to the low incidence of DRESS, NNT estimates for HLA testing might vary significantly among studies [147,148]. Konvinse et al. [129] have estimated an NNT of 75 for the HLA-drug pair HLA-B*32:01–vancomycin in European populations, supporting its potential use routinely. However, given that the population of the European Union is 447 million, the annual hospitalisation rate approximately 1/10 [149], that 2% of hospitalised patients are usually exposed to vancomycin [150,151], and that more than 40% of them receive vancomycin for 2 weeks or more [152], more than 4000 DRESS diagnoses due to HLA-B*32:01 should be expected yearly in the European Union, which largely exceeds the annual rate of total drug hypersensitivity reactions reported in the EudraVigilance tool (n = 383 for the year 2022) [151]. Consistently, HLA-B*32:01 testing is currently not included among recommended tests by drug regulatory authorities and pharmacogenetics working groups [150].

#### 2.3.2. Molecular Mechanisms of T-Cell Activation and Aberrant HLA/TcR Interactions

Viral and pharmacological triggers can disrupt physiological HLA–T cell receptor (TcR) interactions through multiple mechanisms (Figure 2), which may also co-occur in the same subject [153]. A first set of mechanisms are supposed to alter peptide presentation by haptenation of self-molecules or by modification of HLA steric properties.

The hapten-carrier model constitutes the simplest pathophysiological mechanism accounting for drug hypersensitivity. In this setting, the culprit drug activates T-cells after binding intracellular proteins, which are subsequently processed and presented by antigen-presenting cells [154]. Drug binding to HLA might also induce conformational changes causing a shift in HLA affinity to self-peptides, which in turn promotes autoreactive responses [155].

Drugs might also bind HLA or the TcR by non-covalent direct pharmacological interaction (p-i concept), prompting T-cell activation [156]. This mechanism has been described with several drugs classically involved in DRESS, such as carbamazepine [157] and allopurinol [10] For a more comprehensive review on the p-i model, the reader is referred to Pichler 2019 [158]. The p-i concept seems relatively more suitable to explain severe reactions to drugs, such as those observed in DRESS, since it is compatible with the activation of different T-cell clonotypes, rather than the one or few expected in the case of the hapten-specific activation model [158]. This scenario is reminiscent of an alloreactive activation, similar to that observed in GvHD reactions [159]. The paucity of HLA variants capable of being engaged in this dangerous liaison with the drug (or its metabolites) could contribute to explain the low frequency of this condition. This consideration needs to be kept in mind in order to explain why only a subgroup of individuals bearing a high-risk HLA allele do actually develop DRESS following intake of the culprit drug [53].

On the other hand, heterologous reactivity of TcR (heterologous immunity) is a well-recognised mechanism accounting for the ability of the relatively limited human repertoire of T-cell clonotypes to respond to a broad variety of pathogens, even after first antigen exposure, and also for natural autoreactivity towards drugs [160,161]. Promiscuous T-cell activation might also lead to cross-reactivity among drugs sharing a similar chemical structure [162] as well as among viral and self-peptides [163], possibly accounting for hypersensitivity in drug-naïve subjects [164]. Viral factors might also play a role in shifting the immune response towards selected T-cell clones prone to drug-induced activation through recurrent reactivations [48,54,63,165]. Consistent with this model, a study by Yerly et al. [166] showed that self-peptides able to bind permissive HLA variants (e.g., HLA-B*57:01 for abacavir hypersensitivity) may show sequence similarity to (herpes) viral peptides, which can in turn promote T-cell activation. In another report, HLA-B*57:01-restricted HIV-specific T-cells proliferated in response to HLA B*57:01-expressing cells in vitro only in the presence of abacavir [167].

Little is known about the role of heterologous immunity in DRESS (Figure 3). Picard et al. showed that drug exposure prompts expansion of CD8+ T-cells sharing the same TcR repertoire of EBV-specific cytotoxic T-cells in DRESS patients [54]. Niu et al. also showed that EBV-specific CD8+ responses correlate with CD8+ plasticity and DRESS severity, suggesting that repeat challenge by viral factors might continuously renew a pool of autoreactive CD8+ T-cells in predisposed individuals [168]. Heterologous immune mechanisms might also account for potential class sensitisation to multiple drugs in the setting of DRESS [27].

#### 2.3.3. T-Cell Polarisation and Functionality

Aberrant antigen processing and T-cell activation in DRESS encompass alterations in T-cell polarisation and functional specialisation. In fact, patients with DRESS are characterised by oligoclonal expansion of lymphocytes expressing defined subsets of TcR [169]. In addition, patients show a Th2/Treg-skewed phenotype characterised by the possible coexistence of defective control of viral stimuli (that is, reactivated viruses) and enhanced eosinophil proliferation and organ infiltration leading to tissue damage. Supranormal expression of CD134 on circulating CD4+ cells has been detected in patients with DRESS and might contribute to the promotion of Th2 responses [120], besides directly facilitating viral spreading (see above). Enhanced systemic expression of the CD194 (CCR4) ligand *Thymus and Activation-Regulated Chemokine* (TARC) constitutes another hallmark of the acute phase of DRESS. Mechanistically, elevated TARC levels might promote tissue infiltration by Th2 and Treg along with HHV-6 reservoir cells such as mono/myeloid precursors [22,170], favouring HHV-6 spreading and replication, through immunosuppressive responses, and eosinophil-driven tissue damage [57,58,171]. Consistently, patients with predominant HHV-6 reactivation present TARC levels significantly higher than those without HHV-6 reactivation, and in the acute stage of DIHS, these levels correlate with disease activity [22,57,58,171]. 

In contrast to the acute phase of the disease, where IL10-producing classical monocytes are increased, patrolling monocytes are mobilised and release high amounts of IL6. This in turn induces a drift towards Th17-dominated responses [172], while inhibiting Treg cells. Treg cells collected from patients with DRESS in the late phase of the disease show impaired ability to inhibit their effector counterpart in comparison to Tregs from healthy subjects and Tregs collected from patients with early-stage DRESS [173]. In this scenario, late reactivation of other herpesviruses usually occurs.

### 2.4. Eosinophils

Deranged eosinophil inflammation is a hallmark of DRESS and contributes to organ damage. Eosinophils take part in the early-phase response against microbial threats, and their defensive capacities are impaired in eosinophil-driven diseases [174,175,176,177]. Besides performing direct viral clearance tasks as granulocytes, eosinophils also contribute to shaping the downstream inflammatory response to a Th2 profile. In this setting, eosinophils may be stimulated by IL5 released from type II innate-like lymphoid cells (ILC2s) following alarmin release from infected/damaged tissues and in turn promote CD4+ T-cell polarisation towards a Th2 profile by enhancing ILC2 activation and T-cell maturation by releasing IL4 [178]. Consistently, activation-prone ILC2s have been shown to increase in the blood and skin lesions of patients with DRESS along with elevated circulating levels of the alarmin thymic stromal lymphopoietin (TSLP), of the alarmin receptor ST2 (which binds IL33, a potent stimulator of ILC2) and of IL5 [179]. Besides primary eosinophil activation following organ damage, IL5-dependent systemic eosinophilic responses may be sustained by T-cell activation (see above) [180]. Elevated levels of other Th2-associated chemokines, including TARC and macrophage-derived chemokine (CCL22), have also been described in patients with DIHS/DRESS [120]. Persistently high eosinophil counts are thought to correlate linearly with the development of organ damage [181] and are associated with an increased likelihood of eosinophil infiltration of non-physiological eosinophil-homing tissues such as the skin, the liver, the myocardium, and peripheral or central nervous fibres [178,182]. Consistently, these tissues are an integral part of the clinical spectrum of DRESS.

### 2.5. Pathophysiological Basis of DRESS Clinical Manifestations

Multiple aspects of DRESS pathophysiology remain obscure. Nonetheless, available evidence globally suggests that viral reactivation, aberrant drug metabolism, and drug–receptor interaction might prime T-cells to activation besides contributing to part of the early tissue/organ damage. T-cell stimulation may then lead to reactivation of viral genomes harboured by leukocytes, which would eventually further stimulate the immune response to control the spread of actively replicating viral particles. Aberrant differentiation of antiviral T-cell precursors, due to heterologous immune mechanisms, might also enhance drug hypersensitivity and promote delayed-type eosinophil responses, exacerbating organ damage. Persisting viral replication after drug clearance might account for slowly resolving symptoms and potential long-term sequelae as observed in DRESS [29,51].

## 3. Clinical Presentation and Laboratory Findings: When to Suspect DRESS

Multiple organs and tissues can be affected by DRESS syndrome. Systemic findings sorted by declining frequency include cutaneous, lymphatic, haematological, and hepatic manifestations, followed by renal, pulmonary, and cardiac involvement. Severe, atypical cases of DRESS may show neurologic, gastrointestinal, and endocrine dysfunction.

### 3.1. Systemic and Laboratory Findings

Systemic symptoms constitute a hallmark of DRESS. Fever develops in 90% of subjects, but body temperature rarely (7% of cases) exceeds 38.5 °C. Lymph-node enlargement may be detected in up to 60% of patients with DRESS. In the absence of peripheral lymphadenopathy, pathological lymph nodes may be relatively more frequently detected in the mediastinum [183]. Activation of the reticuloendothelial system also causes leucocytosis, with white blood cell count exceeding 10,000 cells/µL in the vast majority of patients. Lymphocytosis and detection of atypical lymphocytes (with increased cellular volume due to expanded cytoplasm along with irregularly shaped nuclei) are also common. Elevated eosinophil count is a defining feature of DRESS, and up to 80% of subjects show hypereosinophilia (that is, eosinophil count exceeding 1500 cells/µL). Neutrophilia and monocytosis might also be part of the blood cell count profile of patients with DRESS, while alterations in platelet count have been less frequently reported [15,184,185].

Abnormal erythrocyte morphologies (AEMs) were studied in a cohort of 215 patients: 32 had AEMs (14%). AEMs were more frequent among patients with DRESS than in patients with other skin manifestations. This phenomenon may be due to DRESS-related perturbation of haematopoiesis. In fact, in DRESS, toxic eosinophilic granule proteins are released and could affect bone marrow. The most frequent AEMs found in DRESS patients are poikilocytosis (48% of patients with AEM), polychromasia (48%), burr cells (33%), ovalocytes (33%) and others [186]. Recently, a study found that levels of TNF- α in blood samples could be useful biomarkers to detect HHV-6 infection. Indeed, its levels were higher in the reactivation group and decreased together with C-reactive protein and lactate dehydrogenase after infection resolution.

### 3.2. Cutaneous Manifestations

Cutaneous eruptions are the most common clinical finding in DRESS. Symmetrical maculopapular eruption involving either the trunk or the extremities is present in 15% of patients [187]. According to the RegiSCAR prospective study [15], polymorphous maculopapular rash is the most common presentation (85%) and encompasses findings such as purpura, infiltrated plaques, blisters, and exfoliative dermatitis. Facial oedema is observed in 70% of patients with skin manifestations. Some patients can develop an exfoliative dermatitis. The rash extent is > 50% of the body surface in approximately 70% of patients [15]. Some patients can present mucosal lesions, most commonly oral lesions and cheilitis [188]. Cutaneous eruption may last more than 2 weeks.

### 3.3. Internal Organ Involvement

#### 3.3.1. Liver, Gastrointestinal, and Pancreatic Involvement

Liver is the most common extracutaneous organ involved in DRESS/DIHS and is often completely asymptomatic. Conversely, liver function abnormalities occur in up to 70% of patients. Anicteric hepatitis is more prevalent, but, if icteric hepatitis occurs, the prognosis is usually poorer, with progression to hepatic failure [80,189]. Hepatic necrosis may rarely develop, although more than 10% of cases may progress to death or need for liver transplantation [83,190,191]. Sulphonamides/sulfones pose the highest risk of inducing liver injury in DRESS, followed by antiepileptic drugs and allopurinol. A 2013 retrospective study on 136 patients suggested that antibiotics, especially β-lactams, are the most frequent culprits for liver injury [192]. According to this study, liver injury is more common in DRESS/DIHS than SJS/TEN, and is usually accompanied by renal failure. Indeed, both drugs and herpesviruses are known to cause liver injury. In fact, HHV-6 not only infects lymphocytes but can also show hepatotropism [57,193], and several reports of CMV hepatitis in the context of DRESS/DIHS have been described in the literature [83]. As anticipated, reactivation of HHV-6 has been more commonly observed in patients with severe clinical findings including long-lasting high fever, leucocytosis, renal failure, and severe hepatitis [48]. Viral hepatitis panels are usually negative in DRESS, but when DRESS associates with an underlying viral hepatitis infection, the disease course can be more complicated and severe [194].

Intestinal involvement has been seldom described in association with DRESS. Descamps et al. reported on a 32-year-old man with sulfasalazine-induced DIHS/DRESS and reactivation of HHV-6 associated with colonic infection and subsequent development of Crohn’s disease [195]. Intestinal involvement in DRESS has also been linked to CMV reactivation, in light of CMV tropism for the intestinal mucosa. This should be suspected when patients develop trunk and intestinal ulcers. The diagnosis is confirmed by detecting anti-CMV IgM and increased CMV DNA copies in blood samples. As anticipated, CMV reactivation is observed 4–5 weeks from the beginning of the disease and can be associated with concomitant or previous detection of HHV-6 reactivation [196]. Gastrointestinal haemorrhage caused by CMV has an unpredictable course and may frequently result in death. For this reason, early detection of CMV reactivation is necessary for successful management of DIHS/DRESS patients, and early administration of anti-CMV treatment can keep these patients from developing acute symptoms [56,59]. A case of esophagitis in DRESS syndrome also has been described [197]. 

Very limited evidence exists for pancreatic involvement in DRESS. A 2003 case report described a 40-year-old black woman treated with allopurinol who developed facial oedema, erythroderma and pyrexia, along with pancreatitis and hepatitis. A potential diagnosis of EBV-associated DRESS was hypothesised. However, non-DRESS-related EBV-induced pancreatitis cannot be ruled out, given that EBV infection might independently cause pancreatitis [198]. Some other cases of pancreatic injuries are described, one of them with concomitant development of diabetes mellitus type 1 after carbamazepine treatment and HHV-6 positivity detection [199].

#### 3.3.2. Kidney Involvement

Kidney involvement occurs in 10–35% of patients [14] and usually manifests as acute interstitial nephritis. Acute renal failure occurs in up to 8% of patients, with a minority requiring renal replacement treatments. Patients are usually clinically silent, but some can present with mild haematuria and proteinuria. In blood analysis, elevated blood urea nitrogen and creatinine levels may point towards renal impairment. Eosinophils may be detected on urinalysis. Kidney ultrasound is usually negative [200]. In most cases, there is only mild renal impairment, which usually resolves after withdrawal of the offending drug. Renal involvement in DRESS is more common after allopurinol treatment, followed by carbamazepine and dapsone [201]. Indeed, allopurinol is generally associated with acute kidney injury (AKI), and renal biopsy typically yields an acute interstitial nephritis (AIN). Rarely, allopurinol can elicit renal vasculitis and glomerulonephritis [202]. On the other hand, renal failure in CBZ-DIHS/DRESS is considered to be attributable to acute interstitial nephritis. Acute interstitial nephritis is typically reversible after withdrawal of the causative agent. Viral agents can directly affect the development of renal involvement in DRESS. Hagiya et al. described a case report of granulomatous tubulointerstitial nephritis accompanying the proliferation of HHV-6 in tubular epithelial cells, demonstrating a possible association between reactivation of HHV-6 directly in the renal tissue with tubulointerstitial nephritis and renal dysfunction [203].

#### 3.3.3. Heart and Muscle Involvement

The myocardium represents a preferential site for eosinophilic infiltration. Consistently, heart involvement in DRESS/DIHS syndrome has been described, and myocarditis represents one of the most relevant prognostic factors in DIHS/DRESS patients [29,46,49,59,67,82,83]. Heart disease is described at onset of disease or after approximately 40 days. Most frequent symptoms are tachycardia, chest pain, dyspnoea, and hypotension, although some patients are completely asymptomatic. According to the literature, ampicillin and minocycline are more frequently responsible for this manifestation [194].

Chest radiography shows cardiomegaly and/or pleural effusion. ECG shows ST-T non-specific abnormalities and sometimes arrythmias. Echocardiography shows significant ejection fraction reduction. Moreover, elevation of creatine kinase and troponin T is usually detected [200]. Two forms of myocarditis are recognized in DRESS syndrome: hypersensitivity myocarditis and acute necrotizing eosinophilic myocarditis (ANEM). The first is usually self-limiting. ANEM is associated with >50% mortality and a median survival of 3 to 4 days [194]. Echocardiography in patients with ANEM shows increased wall thickness, severe biventricular failure, and a pericardial effusion. For both clinical entities, endomyocardial biopsy is important for a definite diagnosis, and it helps in differential diagnosis from other myocarditis. Eosinophilic and mixed lymphohistiocytic infiltrate without necrosis is a typical histological finding of eosinophilic myocarditis. ANEM is defined as eosinophilic and lymphocytic infiltrate with associated myocyte necrosis. Moreover, it is largely known that HHV-6 can cause myocarditis. The hypersensitive response to drug metabolites and the reactivation of a virus such as HHV-6 may be the requisite “immune alteration” in certain individuals that leads to the severe damage of myocardial tissue by eosinophilic degranulation in this disease [204].

#### 3.3.4. Lung Involvement

As well as other organs, the lungs may also be involved in DRESS syndrome. Indeed, pulmonary manifestations may be among the first evidence of the syndrome, anticipating even skin manifestations. According to a review of the literature, pulmonary symptoms might be found in up to 72% of patients on hospital admission [183]. Symptoms of pulmonary involvement are generally dyspnoea, cough and pleurisy [194,200,205]. The most common pulmonary findings include infiltrating lesions of an interstitial nature, pneumonia (50%), and pleural effusion (22.7%) [25,205]. In some cases, pulmonary nodulations have been reported. Very severe manifestations of DRESS syndrome in the lungs can also lead to the development of acute respiratory distress syndrome (ARDS) with acute respiratory failure (31% of cases). The most important risk factors for the development of severe pulmonary manifestations of DRESS accompanied by ARDS seem to be an onset latency shorter than 30 days and age less than or equal to 60 years [183]. 

### 3.4. Nervous System Involvement

Neurological manifestations of DRESS might involve the central and peripheral nervous system. Encephalitis and meningoencephalitis usually present in 2–4 weeks from disease diagnosis and encompass symptoms such as coma, seizures, headache, and speech disturbance. Evidence of HHV-6 DNA in the cerebrospinal fluid points to a probabile role of HHV-6 in this setting [206]. Moreover, a possible link between DIHS/DRESS with reactivation of EBV and the development of autoimmune limbic encephalitis has been described [207]. Electroencephalography shows diffuse slow waves with an occasional solitary spike and waves in the left frontal and temporal leads without periodic patterns. MRI shows bilateral hyperintensity of the grey matter involving the amygdala, medial temporal lobe, insula, and cingulate gyrus. Peripheral involvement is anecdotal [208].

### 3.5. Other Manifestations

An increased prevalence of autoimmune diseases has consistently been observed in DRESS/DIHS survivors. While viral triggers such as EBV and HHV-6 are known potential risk factors for the development of autoimmune diseases such as type 1 diabetes mellitus and autoimmune hypothyroidism [69,209] in the general population, increased lymphocyte counts along with late-phase hypergammaglobulinemia (in contrast to the early phase of DRESS: see above), low levels of interleukin (IL)-2 and IL-4 at DRESS onset and severe liver involvement might synergise with persistent reactivation of EBV and HHV-6 in enhancing post-DRESS autoimmunity [55,210]. Unstable CD8+ T cell repertoires, possibly due to viral stimuli, might also selectively associate with autoimmunity [168].

## 4. Diagnosis

### 4.1. Diagnostic Approach

DRESS should be highly suspected in patients who have recently started new treatments and present with cutaneous eruptions, fever, hypereosinophilia, and alterations in organ function tests [56,211]. Assessment of the causative drug and of the starting time of therapy is one of the first steps of the diagnostic approach when DHR is suspected. 

Initial laboratory investigations are aimed at confirming DRESS diagnosis and evaluating the degree of severity of organ involvement. Laboratory tests include complete blood count with peripheral blood smear for evaluation of eosinophilia (>700/µL), leucocytosis, and the presence of atypical lymphocytes. Significant liver function test abnormalities in more than two measurements are suggestive of liver involvement. Kidney function test abnormalities, proteinuria >1 g/day, or haematuria are suggestive of renal involvement. Cardiac enzymes such as troponins, creatine kinase-MB, and NT-proBNP and pancreatic enzymes such as amylase and lipase should be measured as clinically appropriate [21]. Comprehensive serial laboratory investigations are recommended during the follow-up [21,56].

Reactivation of HHV-6 or other herpesviruses could be assessed by serology or viral genome testing by polymerase chain reaction (PCR) in blood or other tissues [29,56,70,171,190]. There is no universal consensus on the methods to assess viral reactivation, and heterogeneity among different laboratories is the rule [212]. Screening for acute viral hepatitis (e.g., anti-hepatitis A, anti-hepatitis B surface antigen, anti-hepatitis B core antigen IgM, or hepatitis C viral RNA) could be performed to exclude alternative diagnoses in patients with abnormal liver enzymes. The Monospot test is often used as a stand-alone evaluation of infection, despite its low clinical value [213]. Additional tests such as blood cultures or anti-chlamydia, anti-mycoplasma or antinuclear antibodies could be considered for differential diagnosis with other infectious or autoimmune diseases [21,48,54,56,212]. Imaging with ultrasound, computed tomography, echocardiography and cardiac magnetic resonance can be performed to assess the severity of organ involvement. Growing evidence supports the use of ECG and echocardiograms to screen for cardiac manifestations of DRESS, which may have a fulminant course [214].

Patch testing may be useful to ascertain culprit drugs. Recent studies report patch test positivity in 30–60% of patients with DRESS, especially in those cases caused by carbamazepine, β-blockers and PPI administration. Negative results were reported when testing for allopurinol and sulfasalazine [215]. Intradermal testing should be used only in exceptional cases, as there is a risk of reaction recurrence. Drug challenge is contraindicated, although it may be useful in the setting of multiple drug treatment for HIV infection or tuberculosis [211]. In addition, the lymphocyte transformation test (LTT) measures T-cell proliferation, following in vitro exposure to the causative drug [216]. For diagnostic purposes, this assay, which is not available to the routine clinical lab, is best performed in the recovery phase of DRESS. Positive LTT is expected to be found in half DRESS cases [216]. Consistently, LTT sensitivity and specificity are quite high( 73% and 82%, respectively).

Skin biopsy may provide further evidence supporting DRESS diagnosis, although no histopathological finding is pathognomonic. Histopathologic examination may help to rule out other diagnoses such as exanthematous drug eruptions, acute generalized exanthematous pustulosis (AGEP), and Stevens–Johnson’s syndrome/toxic epidermal necrolysis (SJS/TEN). The main histopathological findings encompass dyskeratosis (53–97% prevalence), interface vacuolization (74–91% prevalence), spongiosis (40–78% prevalence), perivascular lymphocytic and dermal eosinophil infiltrates (prevalence 20–80%) [217]. Wide areas of keratinocyte necrosis are found in severe cases [217]. Biopsy of other commonly involved organs (lymph nodes, kidney, liver, and heart) are not routinely performed due to highly nonspecific inflammatory patterns.

### 4.2. Differential Diagnosis

Due to the heterogeneity of its clinical presentation, DRESS syndrome can be misdiagnosed. The main differential diagnoses include acute generalized exanthematous pustulosis (AGEP), exanthematous drug eruptions and Stevens–Johnson syndrome/toxic epidermal necrolysis (SJS/TEN). SJS/TEN shares with DRESS causative drugs, fever, haematological abnormalities and hepatic involvement. While SJS/TEN generally presents a latency period ranging from a few days to 3 weeks, DRESS is a late severe reaction to pharmacological exposure taking at least 2 weeks to show symptoms. On the other hand, AGEP is usually characterised by rapid onset of skin reaction. Referring to hepatic involvement, while more frequent in DISH/DRESS, noteless elevations up to three times the normal value in serum aminotransferase can be detected in half TEN patients (10% of TEN patients can also develop full-blown hepatitis) [218]. Exfoliative dermatitis differs from DRESS in terms of cutaneous and other systemic features. In particular, AGEP rarely associates with kidney involvement and, when present, it is usually self-limited. In addition, visceral involvement in AGEP has rarely been described. In SJS/TEN, a serum urea level >10 mmol/l is a poor prognosis marker [219]. In terms of skin manifestations, DIHS is characterized from the very beginning by a maculopapular rash, often accompanied with oedema on the face and limbs. These features can later evolve to erythroderma or exfoliative dermatitis. In contrast with SJS/TEN, there is no haemorrhagic mucocutaneous involvement. Moreover, HHV-6 re-activation is commonly observed in patients with DIHS/DRESS, along with atypical lymphocytes, whereas it is rarely found in SJS/TEN [220]. AGEP is characterized by an erythematous rash with pustulosis (non-follicular, sterile pustules < 5 mm in diameter) accompanied by fever and neutrophilia and develops typically within 48 h to 3 weeks after drug ingestion. Skin lesions typically observed in AGEP are usually self-limited desquamative pustules accompanied by erythema, with typical facial and intertriginous distribution/pattern and mucosal involvement. Skin biopsies demonstrate intraepidermal pustules with oedema of the papillary dermis and perivascular infiltrates of neutrophils and eosinophils [221,222] (Table 3).

The diagnosis of DRESS may also be particularly challenging in the case of prominent lung involvement because respiratory symptoms associated with peripheral eosinophilia and rash can be found in different pathologies, both infectious and non-infectious (neoplasm, drug, allergic, autoimmune). Specifically autoimmune diseases, acute eosinophilic pneumonia, eosinophilic granulomatosis with polyangiitis [223], idiopathic hyper-eosinophilic syndrome and systemic lupus erythematosus [224], are particularly difficult to distinguish from DRESS. As for infectious causes, all viral, bacterial, parasitic, and fungal pathogens can mimic the symptomatology. The presence or absence of a certain symptom or sign, and a careful anamnesis, help in making a differential diagnosis. In HIV-immunocompromised individuals, a possible cause of DRESS can be found in patients taking raltegravir, whose manifestations characteristically occur in the lungs [225]. The involvement of other districts in addition to the lungs helps to make a differential diagnosis, e.g., renal co-involvement makes one consider pulmonary-renal syndrome (Goodpasture’s), hepatic involvement, and hepatopulmonary amebiasis (Entamoeba histolytica) [183,226]. As the differential diagnosis is quite complex, several studies have shown that at least 50% of patients with DRESS syndrome, given misdiagnosis, were initially treated with antibiotics on suspicion of infection [205,227]. The first step, which is particularly important, is to rule out an infectious aetiology, since corticosteroids, as the mainstay of therapy for DRESS, might instead promote infection and, therefore, be contraindicated. Another condition commonly misdiagnosed with DRESS is lymphoma, for which DRESS is misdiagnosed in up to 75% of cases [14,227]. The presence of characteristic interstitial lung lesions, fever, and dyspnoea may point toward a diagnosis of acute eosinophilic pneumonia, which certainly needs to be differentially diagnosed. In the case of suspected eosinophilic pulmonary pathology, although not a practice that can necessarily be used to make a diagnosis of DRESS, it is useful to perform bronchoscopy in order to collect broncho-lavage (BAL) specimens. Indeed, in the case of differential diagnosis with acute eosinophilic pneumonia (AEP), there is evidence of eosinophilia in BAL samples, in the absence of peripheral eosinophilia, which on the other hand is abundantly present in DRESS. AEP laboratory samples usually show neutrophilic leucocytosis without hypereosinophilia. In the case of eosinophilic pneumonias (EPs), the manifestations may also be secondary to exposure to toxins and drugs. A recent literature review identified 196 cases of drug-induced EP over a 27-year period, with a higher prevalence of AEP than the chronic form. In this case, eosinophilia on peripheral blood was elevated, with mean values from 1232 to 1490 cells/µL in acute and chronic forms, respectively. Compared with the more common forms of EP, drug-induced forms of EP have eosinophilic leucocytosis on blood, as opposed to the far more common neutrophilic form. However, in both the drug-induced and non-drug-induced forms, eosinophilia on BAL is present [228]. Although EPs generally present with low blood eosinophilia, those secondary to drug exposure may instead exhibit hypereosinophilia. In these cases, the differential diagnosis usually relies on cell count in BAL samples, which will show increased eosinophils in almost all cases of EP and lower cellularity in cases of DRESS.

### 4.3. Diagnostic and Prognostic Scoring Systems

Several scoring systems have been suggested over the years to guide the diagnosis of DRESS. Currently three major scoring systems are available, including Bocquet’s published in 2006, the Japanese Consensus Group Severe Cutaneous Adverse Reactions (J-SCAR) in 2006, and the European Registry of Severe Cutaneous Adverse Reactions (RegiSCAR) in 2007. Relevant items from the three scores are reported in Table 4. The RegiSCAR scoring system is more frequently used and includes seven clinical characteristics. Each characteristic is scored up to a total score in order to identify the diagnosis as possible, probable, or definite [15]. Recently introduced Spanish guidelines for DRESS suggest the use of RegiSCAR criteria for clinical diagnosis [23]. Indeed, they provide specific laboratory investigations that are necessary to guide the diagnostic process. In a comparative retrospective analysis conducted by Kim et al., Bocquet’s criteria resulted in being the easiest to be applied in clinical practice [229], although they did not provide specific parameters to potentially guide the differential diagnostic process. A recent review by Cardones et al. compared the abovementioned scoring systems. Interestingly, the Japanese consensus group expanded Bocquet’s criteria, adding more detailed requirements based on clinical manifestations. Differently from Bocquet’s and RegiSCAR criteria, the Japanese consensus group also included HHV-6 reactivation among the relevant items for the diagnosis. Nonetheless, DIHS diagnosis through the J-SCAR criteria is largely consistent with probable/definite DRESS diagnosis according to the RegiSCAR algorithm. This finding confirms that DRESS and DIHS are part of the same heterogeneous disease [230]. One of the major advantages of using the RegiSCAR diagnostic criteria is the possibility to simultaneously differentiate DRESS from other, similar conditions such as acute cutaneous lupus and connective tissue diseases through autoantibody testing or skin biopsy. Ruling out infections through screening for viral acute hepatitis, blood culture, and testing for atypical bacteria such as Chlamydia and Mycoplasma is also part of the RegiSCAR algorithm [15,231].

Despite evidence suggesting the use of RegiSCAR as a complementary diagnostic tool, some limitations might be highlighted. For example, prolonged resolution time (>15 days) is one of the diagnostic criteria but has little usefulness for early diagnosis. Furthermore, assessment of viral reactivation is not included in the scoring system, even though it is considered as a marker of severe disease progression [23,230]. Interestingly, there is increasing effort in identifying CMV disease as a potential prognostic factor for disease severity in patients with DiHS/DRESS. Mizukawa et al. conducted a retrospective analysis of patients with DiHS/DRESS aimed at developing a composite score based on demographic data, medical history and clinical data to monitor severity at different stages of disease, predict patient prognosis, and stratify the risk of CMV disease and complications. Although the study was conducted on a limited number of patients, the composite scoring system was useful in predicting complications related to CMV and to guide medical decisions for early intervention in those patients considered at high risk of CMV reactivation [232].

## 5. Management

The management of patients with DRESS is generally in-patient, although in milder cases (patients without visceral involvement) outpatient management may be considered with clinical and laboratory monitoring every 48 h [23]. The first necessary measure in the case of DRESS is discontinuing potential culprit drugs, along with supportive treatments such as fluid integration and antipyretics. Empirical use of NSAIDs and antibiotics in the acute phase is not recommended, because it could trigger DRESS exacerbations [194].

Addition of systemic steroids is usually necessary [55,233], although evidence from controlled trials is lacking. The rationale for corticosteroid use is the anti-inflammatory and immunosuppressive effect through inhibition of activated cytotoxic T-cells and cytokine production [234]. Immunosuppression poses the risk of favouring viral reactivation, especially of late-phase viruses such as CMV. Therefore, the timing and aggressiveness of immunosuppressive treatments (including corticosteroids) should be evaluated on a case by case basis with careful risk weighting [235]. Nonetheless, timely initiation of corticosteroid treatment is crucial to break the core pathogenic loop of the disease, that is, the reciprocal stimulation of viral replication and T-cell responses [233]. Early aggressive corticosteroid treatment appear to contrast both T-cell activation and HHV-6 replication, while delayed or low-dose treatments are thought to have limited impact on T-cell behaviour and HHV-6 viremia [46]. Similar to HHV-6, EBV-DNA loads are significantly debulked by systemic corticosteroids. Furthermore, since EBV is a known trigger of autoimmunity, early systemic corticosteroid treatment possibly prompting reduced EBV-DNA loads might have a fundamental role in optimising patient outcomes especially in settings at high-risk for autoimmunity [56]. It is recommended to start with a minimum dosage of 1 mg/kg/day of prednisone or equivalent, with a taper in 3–6 months [235]. In cases where this is not sufficient, pulse intravenous methylprednisolone may be used.

In refractory cases or when steroids are contraindicated, some studies recommend the use of cyclosporine, due its effects on cytotoxic T-lymphocyte activation and on IL5 inhibition [23,236]. The efficacy of intravenous Immunoglobulins (IVIGs) may be due to their general anti-inflammatory effect, protective effect against herpes virus reactivation and compensation for decreased immunoglobulin levels observed in DRESS [237]. IVIGs also constitute an interesting treatment option as a steroid-sparing agent in cases of concomitant infection [237], but they are not indicated in monotherapy due to potential lack of efficacy and increased adverse event rates [194]. Limited data suggest the potential use of immunosuppressants (cyclophosphamide, mycophenolate, rituximab) or plasmapheresis in refractory cases [235].

CMV reactivation might be integral to the pathophysiology of DIHS/DRESS or constitute an unwanted side-effect of immunosuppression [83]. Since CMV reactivation is among the most important risk factors in the prognosis of DIHS/DRESS, caution in the use of corticosteroids is recommended [46]. Indeed, fatal outcomes possibly related to the use of DRESS-related treatments are found almost exclusively in CMV cases. Delayed anti-CMV therapy is associated with a higher risk of adverse outcomes as patients receiving treatment after three or more days from CMV reactivation detection bear a significantly higher risk of death even compared to patients with treatment start after two days from CMV reactivation detection. Anti-CMV therapy may also have a synergic role in minimising the risk of CMV-related and unrelated pathological events, including other herpesvirus-related complications [56]. Some authors suggest that prophylactic treatment of CMV during viraemic stages could prevent progression to CMV-related clinical manifestations in patients with DIHS/DRESS [23]. Antivirals (ganciclovir or valganciclovir) in addition to standard treatment are usually administered with continuous monitoring of viral loads.

Given the possible systemic involvement, multidisciplinary management may be necessary depending on the organs involved. In cases of exfoliative dermatitis, treatment is similar to that of major burns, and management in burn units should be considered [238]. Generally, most patients respond to treatment; however, it must be remembered that this condition has an estimated mortality of up to 20%, while other subjects may have long-term adverse effects [238]. Patients with DRESS may experience relapses of symptoms in the recovery phase. These events are typically associated with corticosteroid dose reduction and viral reactivation. Interestingly, exposure to new drugs during DRESS could represent an underlying mechanism for relapse occurrence with or without subsequent drug sensitization [239,240].

## 6. Final Remarks

DRESS is a severe multi-organ syndrome characterised by abnormal T-cell and eosinophil responses to drugs along with abnormal control of viral stimuli. Advances in recent years, possibly boosted by the ongoing COVID-19 pandemic and its sequelae for individuals and public health, are increasingly highlighting the role of viruses in modulating the course and severity of DRESS. Large multicentre studies addressing changes in epidemiology and clinical presentation of DRESS among distinct healthcare and microbiological settings are eagerly needed along with deeper mechanistic insights into the pathophysiological basis of DRESS-related immune dysfunction.

## Figures and Tables

**Figure 1 microorganisms-11-00346-f001:**
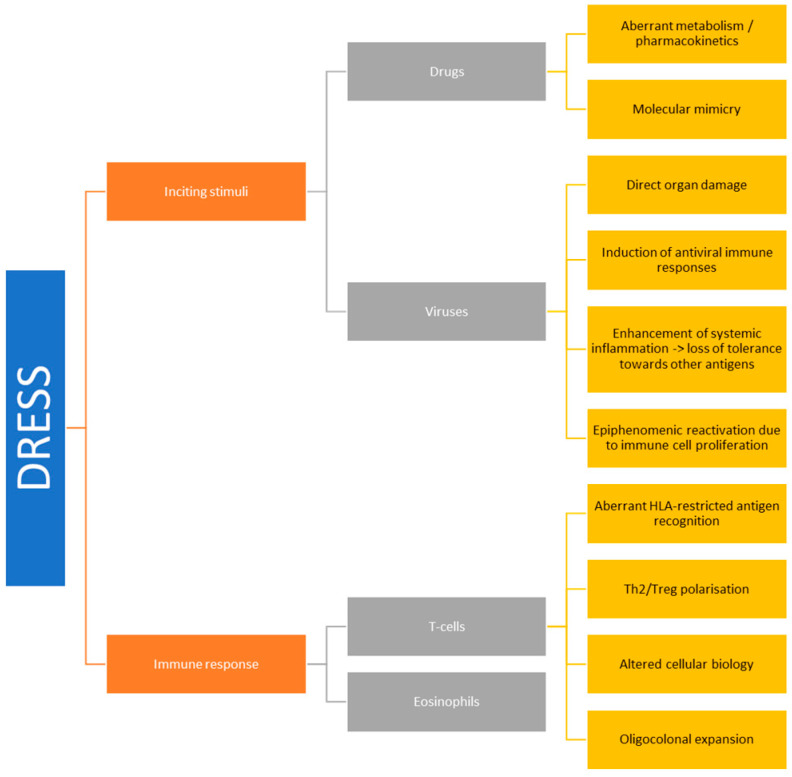
Simplified DRESS aetiopathogenesis. Flow-chart depicting the pathophysiological relationships among the main exogenous and host-related factors involved in the development of DRESS.

**Figure 2 microorganisms-11-00346-f002:**
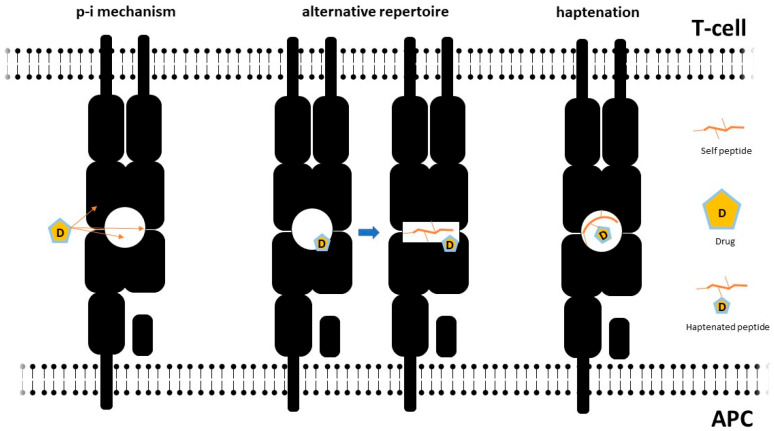
Pathogenic mechanisms for HLA-restricted, drug-induced activation of T-cells in DRESS. In this simplified depiction of HLA-restricted antigen presenting cell (APC)–T-cell interactions, the three main hypothesised mechanisms accounting for drug-induced T-cell activation in DRESS are represented. Drugs might directly interfere with HLA–T-cell receptor interactions, causing T-cell activation without the need for self peptides (p-i mechanism, **left** side). In this setting, drugs might activate the T-cell receptor through allosteric mechanisms or by binding HLA either inside or outside the peptide binding groove. Drugs bound to the peptide groove might cause conformational changes enabling self-peptides to be accommodated within HLA and presented to T-cells, promoting self-reactive responses (alternative peptide repertoire hypothesis, central section). Drugs can also bind self molecules through conventional hapten-carrier models (**right** side).

**Figure 3 microorganisms-11-00346-f003:**
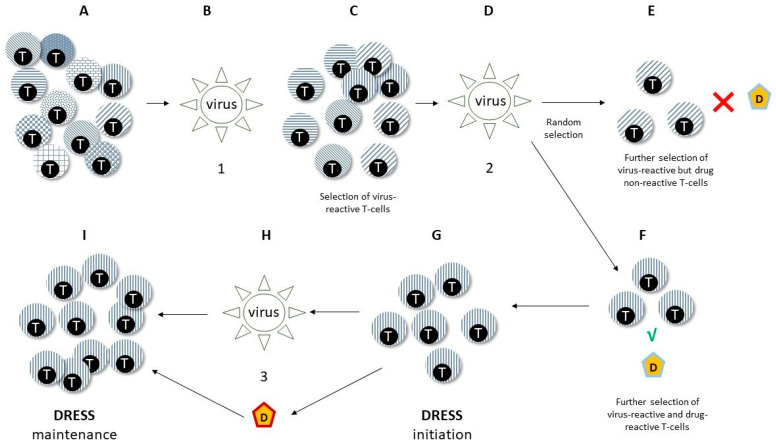
Potential mechanisms of heterologous immunity in DRESS. This figure depicts features of heterologous immunity with a potential pathogenic role in DRESS. Viral infections or reactivations (1–3) prompt selective pressure on a heterogeneous pool of T-cells (**A**). Therefore, after exposure to viruses (**B**), virus-reactive T-cells expand (**C**) and are readily available for eventual viral encounters or reactivations (**D**). Among virus-reactive T-cells, subpopulations harbouring T-cell receptors devoid of the ability to be activated by potential drug allergens might be selected (**E**), preventing the occurrence of hypersensitivity. In other cases (**F**), either occurring in distinct subjects or in the same subjects during distinct phases of life, virus–drug cross-reactive T-lymphocytes might be selected by viral stimulation. When challenged with culprit drugs, these cells might initiate hypersensitivity reactions, possibly including DRESS (**G**). Eventually, re-challenge with re-activating viruses (**H**, **top**) or chemically related drugs (**H**, **bottom**) might promote DRESS progression and/or persistence (**I**).

**Table 2 microorganisms-11-00346-t002:** Selected HLA haplotypes associated with DRESS.

HLA	Viral Infection	Effects on Viral Infection	Drugs	Population	OR (95% C.I.) Ref.
HLA-A*24:02			Lamotrigine	Spanish	34.5 (2.03–209.71) [131]
HLA-A*31:01			Carbamazepine	Han Chinese	12.9 (3.7–45.3) [132]
				Japanese	[133]
				European	24.1 (9.6–60.3) [134]
				North African	32.0 (2.6–389.2) [133]
			Lamotrigine	Korean	11.43 (1.95–59.77) ^§^ [135]
HLA-A*32:01			Vancomycin	European	[129]
HLA-A*33:03			Allopurinol	Korean	25.2 (5.2–121.8) [136]
HLA-B*13:01			Dapsone	Han Chinese	[137]
				Thai	60.75 (7.44–496.18) [138]
				Taiwanese, Malaysian	49.64 (5.89–418.13) [137]
			Sulfasalazine	Han Chinese	11.16 (1.98–62.85) [139]
			Sulphamethoxazole	Asian	61 (21.5–175) [140]
HLA-B14:02			Nevirapine	Caucasian	
HLA-B* 51:01			Carbamazepine	Han Chinese	4.6 (2.0–10.5) [132]
			Phenytoin	Thai	5.2 (1.2–22.7) [141]
HLA-B*53:01	HIV		Raltegravir	African	[114]
HLA-B*56:02			Phenytoin	Australian Aboriginal	[142]
HLA-B*58:01			Allopurinol	Han Chinese, Thai, Japanese, Korean, European	580.3 (34.4–9780.9) ^§^ [10]
			Carbamazepine	Asian	7.55 (1.20–47.58) [143]
	CMV	Increased reactivation risk			
HLA-B*15:13			Phenytoin	Malaysian	59.0 (2.5–1395.7) [144]
HLA-C*03:02			Allopurinol	Korean	135.7 (15.6–1177.8) [136]
HLA-C*04:01			Nevirapine	Malawian	2.6 (1.1–2.6) ^§^ [145]
HLA-DRB1*15:01			IL-1 and IL-6 inhibitors	European patients with AOSD or JIA	40.8 (5.3–316) [39]
	EBV	Coreceptor to EBV infection on B cells			[146]

^§^ Odds ratios available only for DRESS + Stevens-Johnson’s Syndrome + Toxic epidermal necrolysis cumulated in the original paper; AOSD—Adult-onset Still's disease; IJA—Juvenile idiopathic arthritis.

**Table 3 microorganisms-11-00346-t003:** Differential skin manifestations among DRESS and other diseases.

Syndrome	Rash Features	Timing of Onset	Disease Extent	Systemic Manifestations	Other than Skin Involvement	Blood Analysis Findings	Histopathological Findings
DIHS/DRESS	Maculopapular exanthem	2–8 weeks	Generalised	Fever	Hepatitis Lymphadenopathy Pneumonitis Nephritis	Eosinophilia, atypical lymphocytes, leucocytosis	Subtle, vacuolar interface dermatitis, with scattered, dyskeratotic keratinocytes along the dermo-epidermal junction zone
	Erythroderma		Mucosal involvement Rare			Abnormal liver and renal function tests	
	Facial oedema						
AGEP	Generalised erythema	<3 days	Generalised usually with skin fold and facial localisation,	Higher fever (>38 °C)	Rare	Leucocytosis with neutrophilia (>7000/mm^3^)	Intraepidermal pustules with oedema of the papillary dermis and perivascular infiltrates of neutrophils and eosinophils
	Pustules		Mucosal involvement rare				
	Erythroderma						
SJS/TEN	Dusky red, coalescent macular exanthem	4–21 days	Disseminated	Fever Photophobia Sore throat, Dysphagia	Pneumonitis	Lymphopenia	Necrosis of keratinocytes Epidermis shedding Absent inflammatory infiltrate
	Atypical target lesions						
	Bullous lesions		Mucosal involvement rarely absent (stomatitis, conjunctivitis)				
	Epidermal necrosis						
	Nikolsky sign						

**Table 4 microorganisms-11-00346-t004:** Comparison between three scoring systems.

Bocquet et al., 1996	J-SCAR, 2006	RegiSCAR, 2007
Cutaneous drug eruption	Fever	Fever > 38.5 °C
Systemic involvement: lymphadenopathy ≥ 2 cm; liver involvement (transaminase twice the upper limit); kidney involvement (e.g., interstitial nephritis); lung and cardiac involvement (e.g., interstitial pneumonitis or myocarditis)	Latency time of 3 weeks from drug exposure to the onset of cutaneous manifestations	Enlarged lymph nodes in ≥2 lymph node stations
Hematologic alterations: eosinophilia ≥ 1.5 × 10^9^/L; presence of atypical lymphocytes	Persistence of the eruption ≥ 2 weeks after drug interruption	Eosinophilia > 700/µL
		Thrombocytopenia
	HHV-6 reactivation at PCR or serology tests	Atypical lymphocytes
		Skin involvement (rash extended for > 50% of body surface area, biopsy)
		Organ involvement (one or ≥ two organs involved)
		Resolution in ≥15 days
		≥3 negative laboratory investigations including ANA screening, serological screening for HAV/HBV/HCV, blood cultures, tests for Chlamydia and Mycoplasma to exclude other diseases

## Data Availability

Not applicable.

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
