# Peer review of "Drug Reaction with Eosinophilia and Systemic Symptoms (DRESS): Focus on the Pathophysiological and Diagnostic Role of Viruses"

_microorganisms, 2023, doi:10.3390/microorganisms11020346_

Round 1

Reviewer 2 Report

The authors summarize recent important advances made in our understanding of DIHS/DRESS. They focused on the role of viral reactivation in the pathogenesis. Unfortunately, however, the problem is that the authors are not citing original work on several important issues but secondary or tertiary imitated work that they have cited and that the readers will not understand the essence of the work because of the shredded quotations.

Major problems:

1.     Decreased IgG production and decreased B cell counts in DIHS/DRESS are not Drago’s original work.28

2.     Sequential occurrence of herpesviruses reactivation in DIHS/DRESS is not Anti’s original work.59

3.     Dysfunctional Treg cells in the subacute stage is not Hanafusa’s original work.130

4.     A shift to Th 17-dominated response in the resolution stage via monocyte-derived IL-6 is not Cho’s original work.21

5.     Persistent reactivation of EBV and HHV-6 as risk factors of autoimmune sequelae is not Gentile’s,24 Hama’s,55 and Tsutsumi’s original work.

Citing such wrong papers reduced the value of the original work, is a disservice to it, and will lead the readers to make false assumptions. Probably the authors of this Review did not actually follow recent advances in the disease but only examined recent papers that only cited the original work and wrote this review article. Because of the shredded quotes, it lacks the logic that the original work had.

6.     The most important drawback of this Review article was the lack of quotation of the most important article in scoring system, in which the importance of earlier anti-CMV therapy in the prognosis of DIHS/DRESS has been described. This finding was just cited without any citation in this manuscript. This is also problematic.

7.     Although the authors describe awareness of the multi-facet aspects of immune perturbation caused by SARS-CoV-2 infection, inflammatory profiles of post-acute COVID-19 syndrome similar to DIHS/DRESS has not been described in detail.
